# Impact of Different Air Pollutants (PM_10_, PM_2.5_, NO_2_, and Bacterial Aerosols) on COVID-19 Cases in Gliwice, Southern Poland

**DOI:** 10.3390/ijerph192114181

**Published:** 2022-10-30

**Authors:** Ewa Brągoszewska, Anna Mainka

**Affiliations:** 1Department of Technologies and Installations for Waste Management, Faculty of Energy and Environmental Engineering, Silesian University of Technology, 18 Konarskiego St., 44-100 Gliwice, Poland; 2Department of Air Protection, Silesian University of Technology, 22B Konarskiego St., 44-100 Gliwice, Poland

**Keywords:** air pollution, PM_2.5_, PM_10_, NO_2_, bioaerosols, COVID-19, meteorological parameters, atmospheric air, human health

## Abstract

Many studies have shown that air pollution may be closely associated with increased morbidity and mortality due to COVID-19. It has been observed that exposure to air pollution leads to reduced immune response, thereby facilitating viral penetration and replication. In our study, we combined information on confirmed COVID-19 daily new cases (DNCs) in one of the most polluted regions in the European Union (EU) with air-quality monitoring data, including meteorological parameters (temperature, relative humidity, atmospheric pressure, wind speed, and direction) and concentrations of particulate matter (PM_10_ and PM_2.5_), sulfur dioxide (SO_2_), nitrogen oxides (NO and NO_2_), ozone (O_3_), and carbon monoxide (CO). Additionally, the relationship between bacterial aerosol (BA) concentration and COVID-19 spread was analyzed. We confirmed a significant positive correlation (*p* < 0.05) between NO_2_ concentrations and numbers of confirmed DNCs and observed positive correlations (*p* < 0.05) between BA concentrations and DNCs, which may point to coronavirus air transmission by surface deposits on bioaerosol particles. In addition, wind direction information was used to show that the highest numbers of DNCs were associated with the dominant wind directions in the region (southern and southwestern parts).

## 1. Introduction

Respiratory infections are the leading cause of epidemics, causing about 5 million deaths per year around the world [1]. In 2020, we became participants in an unprecedented international public health challenge. As a result of the coronavirus-associated acute respiratory syndrome (SARS-CoV-2), with COVID-19 disease as a symptom, both educational and commercial systems, as well as the general well-being of societies, have suffered [2]. In Poland, the first case of SARS-CoV-2 infection was diagnosed on March 4, while on March 12 the WHO regional director for Europe identified the region as the center of the pandemic. On March 17, every country in Europe had at least one confirmed case of COVID-19 [3].

Poland is a country with one of the largest air pollution problems in the European Union (EU). The Silesia voivodeship is the most polluted region in Poland, a country with 36 cities in a ranking of the 50 most polluted cities in the EU [4]. It has been estimated that, due to exposure to air pollution, the life expectancy of the average Polish citizen is shortened by around nine months, and 48,000 people die prematurely every year due to air pollution [5]. Epidemiological data and pathophysiological mechanisms suggest that ambient air pollution affects both the spread of COVID-19 disease and its severity [6]. Therefore, it is crucial to define the role that air pollutants play in the increase in morbidity and mortality due to COVID-19 [7].

Air pollution is known to damage many organs and systems of the human body. Of particular importance is the reduction in immunity to bacterial or viral infections of the respiratory system and the effect on the functioning of the cardiovascular system [8]. Moreover, after the outbreak of COVID-19, people are infected more often by SARS-CoV-2 in areas with high levels of air pollution than in less polluted areas. Air contamination disables airway mucosal functioning, including the production of fluid that lines the airway surface and contains respiratory host defense peptides, as well as mucus production and the tight junctions between epithelial cells. That is why air pollution can provoke cilia dysfunction, with changed surfactant composition and higher permeability of the airway epithelium [9,10,11]. Consequently, impairment of the mucosal barrier impairs lung defense against inhaled pathogens, such as SARS-CoV-2 [12].

Studies over the past two years on regions with high levels of air pollution have shown correlations with COVID-19 mortality. Regions in Northern Italy, including Lombardy, Veneto, and Emilia-Romagna, can be used as examples [13]. Similar trends have been observed in other regions with high air pollution, such as the Wuhan region of China and the United States, where poor air quality is correlated with a high incidence of COVID-19 and positive results of COVID-19 tests [10]. Currently, research has shown a relationship between routinely measured air pollutants, for example, particulate matter (PM_10_ and PM_2.5_) and nitrogen oxides (NO_x_), and increased numbers of COVID-19 cases [13,14,15,16,17,18,19]. The impact of meteorological conditions has also been analyzed [20,21,22,23,24]. However, there is still a lack of reports on the relationship between bacterial aerosols (BAs) in ambient air and incidence of COVID-19. Worldwide studies have revealed that BA concentrations vary among different types of outdoor environments, with considerable seasonal variations as well [25,26,27]. BA concentration and composition in outdoor air can be influenced by specific micro- and macroscale determinants, such as land use, emission sources, air humidity, temperature, and UV radiation [26,28,29].

Bacterial aerosol particles are significant health risk factors, and exposure to these particles is associated with a varied range of health effects, including three major groups: infections, toxic reactions, and allergic reactions [30,31]. Therefore, the main aim of our study was to determine the impact of bacterial aerosols present in ambient air on the increase in COVID-19 cases in Gliwice in the Upper Silesia region of Poland, which is one of the most polluted areas in the EU [31].

This research is a contribution to the public debate on whether ambient particles can transport viruses that cause COVID-19 [16]. We believe that increasing knowledge of the relationship between air pollution and the incidence of COVID-19 symptoms can be beneficial in informing public health measures all around the world.

## 2. Materials and Methods

### 2.1. Sampling Sites

The study was carried out in Gliwice (50°17′37.1″ N 18°40′54.9″ E). Gliwice is a typical representative of a city located in the industrial area of Upper Silesia, Poland, with 178.186 thousand occupants (Figure 1). It is a densely populated and highly industrialized region of Poland and is responsible for the highest level of coal production. There are numerous coal-fired power plants, coking plants, and steel mills. Due to high levels of air pollution, the Silesia region has the shortest life expectancy and the highest incidence of premature births as well as genetic birth defects in Poland [32].

### 2.2. Measurements of Ambient Air Pollutants

The ambient air pollutants measured included bacterial aerosols (BAs), PM_2.5_, PM_10_, SO_2_, and NO_x,_ including NO and NO_2_, as well as O_3_ and CO; various meteorological parameters, such as relative humidity (RH), air temperature (t), atmospheric pressure (P), and wind speed and direction, were also measured. All measurements were carried out during March 2021 from Monday to Friday. Additionally, an analysis of the impact of PM_2.5_ and PM_10_ concentrations on COVID-19 daily new cases (DNCs) was conducted during the winter season (from November 2020 to February 2021).

The data on PM_10_ concentrations, as well as all gaseous and meteorological parameters, were gathered by the mobile air quality station for air pollutant emission measurements located at the Silesian University of Technology in Gliwice. The measuring equipment includes continuous automatic certificated monitors for PM_10_/PM_2.5_ particulate matter (Beta Attenuation Monitor BAM1020 Met One Instruments, Inc., Grants Pass, OR, USA), SO_2_ (fluorescence analyzer—T100/API-Teledyne, San Diego, CA, USA), NO_x_ (chemiluminescence analyser—T200/API Teledyne, San Diego, CA, USA), O_3_ (UV absorption analyzer—T400/API-Teledyne, San Diego, CA, USA), and CO (infrared energy absorption analyzer—T300/API-Teledyne, San Diego, CA, USA), as well as the meteorological station (Meteo set WS 500 Lufft, G. Lufft Mess- und Regeltechnik GmbH, Fellbach, Germany). Additionally, data on PM_10_ and PM_2.5_ ambient levels were taken from the air monitoring station nearest to the Silesian University of Technology (at a distance of about 2500 m) at Mewy Street. The monitoring station belongs to the National Inspectorate of Environmental Protection in the Upper Silesia voivodeship [33]. The BA concentrations were measured using an Air Ideal (bioMérieux, France) one-stage impactor with an air flow rate of 100 dm^3^/min, at a height of about 1.5 m to simulate aspiration from the human breathing zone, with the same operational details as in our previous studies [34,35]. Air pollutant levels and meteorological parameters are 24 h averages.

In addition, after a 24 h incubation, single colonies of BAs were passaged on a Biolog Universal Growth Agar (24 h incubation at 37 °C). Characterization of the isolates was performed using Gram staining and cell morphological analysis. In the next step, selected strains were then identified using the Biolog OmniLog system (Biolog, Haward, CA, USA) and a GEN III MicroPlate™, as in our previous research [31,35].

### 2.3. Measurements of SARS-CoV-2 Cases

The official data for SARS-CoV-2 infections in Poland are published daily by the Polish Ministry of Health [36]. All cases are diagnosed as positive based on polymerase chain reaction tests for SARS-CoV-2. We collected the cumulative number of cases for the district of Gliwice in Upper Silesia, Poland, from 23 November 2020 (the first day of available data) up to 31 March 2021. The data on the daily new cases (DNCs) due to COVID-19 were obtained from publicly available databases; hence, ethical approval was not required.

### 2.4. Statistical Analyses

The data were analyzed using Statistica software (TIBCO Software Inc. Palo Alto, CA, USA), version 13.3 for Windows, and a *p*-value < 0.05 was considered statistically significant. To determine whether a small data set (*n* < 50) was normally distributed, two tests were used: the Lilliefors test and the Shapiro–Wilk test. Table 1 presents the results of the normality tests of the random distributions of the measured parameters. Normality was revealed for total bacteria levels, daily new SARS-CoV-2 cases, concentrations of NO_2_ and O_3_, as well as all meteorological parameters, except ambient temperature. In the case of these parameters, linear regression could be used. For the other parameters, Spearman’s rank correlation was used to test whether there was concordance (strength and direction) between the total bacteria levels, ambient air pollutants, meteorological parameters, and SARS-CoV-2 cases. Spearman correlations, not Pearson correlations, were used due to the non-normal distribution of the obtained variables (PM fractions, SO_2_, NO, NO_x_, CO, and temperature) generated from the daily time series data.

## 3. Results and Discussion

### 3.1. Particulate Matter (PM) Concentrations and SARS-CoV-2 Daily New Cases (DNCs)

Long-term chronic exposure to air pollutants might play a significant role in the spread of COVID-19 [37]. In addition, short-term exposure to high levels of ground PM concentrations found in ambient air is associated with reduction in lung function and induction of respiratory symptoms, including cough, shortness of breath, and pain on deep inspiration [38,39]. New systematic reports have emphasized a possible association between the transmission of the virus in exposed populations and the level of PM in the atmosphere. However, confounding effects may be present, such as gender, age, smoking status, and high population density, as potential risk factors for higher morbidity and mortality due to COVID-19 [40,41]. Therefore, caution has to be taken in translating values of conventional indicators, such as PM_2.5_ and PM_10_ levels, into measures of vulnerability to COVID-19.

In our study, we observed a relationship between PM concentrations and daily new cases (DNCs). Figure 2 and Figure 3 present the similarity in the daily course of PM_10_ and PM_2.5_ levels and DNCs during the winter season (from November 2020 to February 2021). The plots are consistent with other results showing a relationship between higher air pollution levels and COVID-19 cases [4,7,16,42].

In Poland, just as in many other countries in Central and Eastern Europe, high levels of two PM fractions (PM_10_ and PM_2.5_) are observed every winter season. This is due to the high share of solid fuels in the primary energy source structure and the large share of low communal emissions [43,44]. However, what interested us and became an inspiration for further research was that we observed that the relationship between PM and DNCs had been weakening since March (spring season, end of the heating season), despite the continuous increase in the number of cases of SARS-CoV-2. Table 1 presents the means, medians, and ranges (min–max) of parameters monitored in March 2021.

Following the newest WHO global air quality guidelines [45], the recommended 24 h concentration of PM_2.5_ is 15 μg/m^3^, that of PM_10_ is 45 μg/m^3^, that of sulfur dioxide (SO_2_) is 40 μg/m^3^, that of nitrogen dioxide (NO_2_) is 25 μg/m^3^, and that of carbon monoxide (CO) is 4 μg/m^3^, while, for ozone (O_3_), the recommended 8 h average concentration is 100 μg/m^3^. All gaseous pollutants were found to be below the level recommended by the WHO, while PM fractions exceeded recommended levels. The highest concentrations of major air pollutants monitored during the selected month were observed for both PM_2.5_ and PM_10_. These two fractions determined overall air quality in March 2021.

Figure 4 shows that the contribution of the air quality index (AQI) during March 2021 was mainly moderate. The correlation matrix (Table 2) for SARS-CoV-2 daily new cases (DNCs), ambient air pollutant concentrations, and bacterial aerosol concentrations during March 2021 suggests that, in moderate ambient air conditions, DNCs are significantly correlated with bacterial aerosols (BAs) and NO_2_. The correlation coefficients (*r*) were 0.903 and 0.724, respectively.

### 3.2. Meteorological Conditions and SARS-CoV-2 Daily New Cases (DNCs)

Table 3 shows the results of a correlation analysis of meteorological parameters and DNCs as well as BAs. Interestingly, the analysis revealed a significant negative correlation between wind direction and DNCs (*r* = −0.477), as well as between BAs and atmospheric pressure. Figure 5 presents a wind rose diagram for March 2021 derived from the monitoring by a mobile air quality station located in Gliwice. The results of the study indicated that the wind in the studied area dominantly blew towards the south and southwest, and the wind speed values were low, in a range from 0.5 to 2.8 m/s (Table 1). As can be seen, the highest numbers of COVID-19 cases correspond to wind directions.

### 3.3. Bacterial Aerosol (BA) Concentrations and SARS-CoV-2 Daily New Cases (DNCs)

To our knowledge, no research has previously been carried out to evaluate the link between concentrations of bacterial aerosols (BAs) in the outdoor air and numbers of cases of SARS-CoV-2. These results seem even more interesting given that, for a 10-year period in Poland, we recorded the maximum average concentration of BAs in the spring season (the time these analyses were conducted) and the lowest in the winter. During winter, extreme conditions, such as decreases in temperature and the heaviest rainfall and snowfall of the year, might contribute to the decrease in BA levels. On the other hand, in the summer, it would seem that the most favorable conditions for the growth of bacteria that we observed decreased BA concentrations. The reason for this decline may be the extremely high temperatures and strong UV radiation from the sun noted at this time.

The median BA concentration was 690 CFU/m^3^ and varied in a range from 410 to 980 CFU/m^3^ (Table 1). Figure 6 shows that the BA concentrations were linked to increased numbers of new SARS-CoV-2 cases. Table 2 presents a matrix of correlation coefficients (*r*) for daily new cases (DNCs) and all variables included in the analysis, which suggests that BA concentrations during March 2021 were highly correlated with DNCs (*r* = 0.903) and that the relationship was linear (*R*^2^ = 0.758).

We suggest that bacterial infections may cause increases in the numbers of COVID-19 patients. However, the collection of respiratory samples from this type of patient is complicated because of the elevated risks associated with aerosol generation procedures. Consequently, bacterial respiratory tract infections are likely to be under-detected in patients hospitalized with COVID-19. There are only a limited number of papers that have reported species identities or sampling times, making it impossible to determine whether patients had bacterial infections at the time of hospital admission [46].

A significant correlation (*p* < 0.05) was found between BA concentrations and SARS-CoV-2 in a hospital in Iran, where the obtained results implied that contact with bioaerosols generated through COVID-19 patients’, healthcare workers’, and visitors’ exhalations in hospital wards may pose a serious health threat, especially to susceptible individuals [47]. Zhou et al. found that bacterial infections (bacteraemia and pneumonia) were more common in fatal COVID-19 cases compared with recovered cases in Wuhan, China [48].

There is a suspicion that pollen bioaerosols can also affect coronavirus survival [49,50]. Considering the summer incidence of coronavirus during June 2022, under suitable environmental conditions, simultaneous or co-exposure to SARS-CoV-2 (via other infected human carriers) and airborne aerosols might promote viral infection. Therefore, we must detect the seasonal patterns of bioaerosols and airborne viruses, including COVID-19, based on environmental factors.

The most commonly isolated bacterial group in our research was that of the Gram-positive rods that form endospores, among which *Bacillus* was the most frequently isolated genus (Table 4).

The spores of *Bacillus* have remarkable resistance to chemical and physical factors. This genus of bacteria is commonly found in soil and water and is a component of the normal flora of the skin and mucous membranes of humans and animals [51]. This result corresponds with our previous findings [26,35] and is common to other studies [52,53,54].

### 3.4. Nitrogen Dioxide (NO_2_) Concentrations and SARS-CoV-2 Daily New Cases (DNCs)

Nitrogen dioxide (NO_2_) is another important air pollutant toxic to human respiratory systems when present at higher concentrations in the atmosphere [55]. In our study, the median NO_2_ concentration was 9.5 ppb (Table 1). Figure 7 shows that NO_2_ concentrations were linked to increased numbers of SARS-CoV-2 cases. The relation was linear (*R*^2^ = 0.597), as shown by regression analysis, and the correlation was high (*r* = 0.724); on the other hand, NO and NO_x_ concentrations (0.477 and 0.595, respectively) were linked to increased numbers of SARS-CoV-2 cases to a lesser extent.

Similar results were found in Wuhan, China, where Li et al. (2020) found a significant linear correlation between SARS-CoV-2 DNCs and NO_2_ concentrations (*R*^2^ = 0.329, *p* < 0.001) [56]. In Spain, Italy, France, and Germany, it was observed that out of 4443 fatalities observed at the beginning of the pandemic of COVID-19, 3487 deaths, accounting for 78% of the total deaths, were confined to areas where NO_2_ pollution was predominant [57].

## 4. Conclusions

Understanding the airborne route of SARS-CoV-2 transmission is essential for infection prevention and control, and improvements in terms of air pollution, lifestyle, and the environment will help to prevent future viral pandemics.

Our study found that the role of ambient air pollution given moderate air quality is largely unknown, necessitating further epidemiological studies. Although the current study was conducted only in Gliwice, Poland, it points to the as yet unrepresented implication that bacterial aerosol (BA) concentrations in the period characterized by moderate air quality were significantly associated with SARS-CoV-2 daily new cases (DNCs).

In conclusion, we think that our analyses of the correlations between bacterial aerosol (BA) concentrations and new COVID-19 cases are foundations for further, wider research. However, the drastic mutational nature of the virus makes it difficult to predict which mechanisms and ecological parameters will affect its growth and prevalence.

## Figures and Tables

**Figure 1 ijerph-19-14181-f001:**
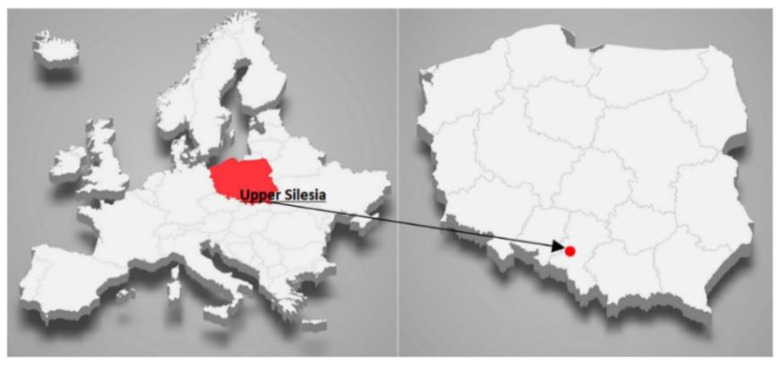
Localization of the measurement point in Gliwice, Upper Silesia, Southern Poland.

**Figure 2 ijerph-19-14181-f002:**
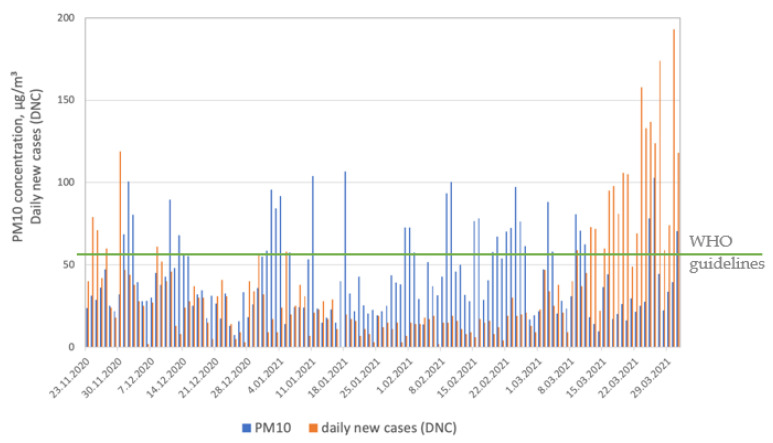
PM_10_ concentrations and SARS-CoV-2 daily new cases (DNCs).

**Figure 3 ijerph-19-14181-f003:**
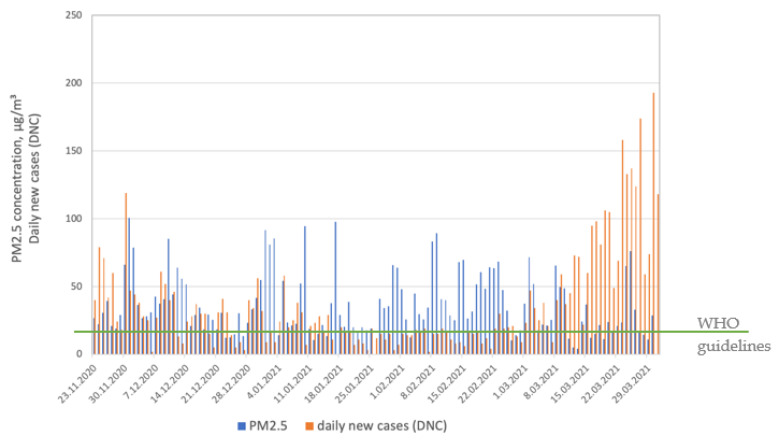
PM_2.5_ concentrations and SARS-CoV-2 daily new cases (DNCs).

**Figure 4 ijerph-19-14181-f004:**
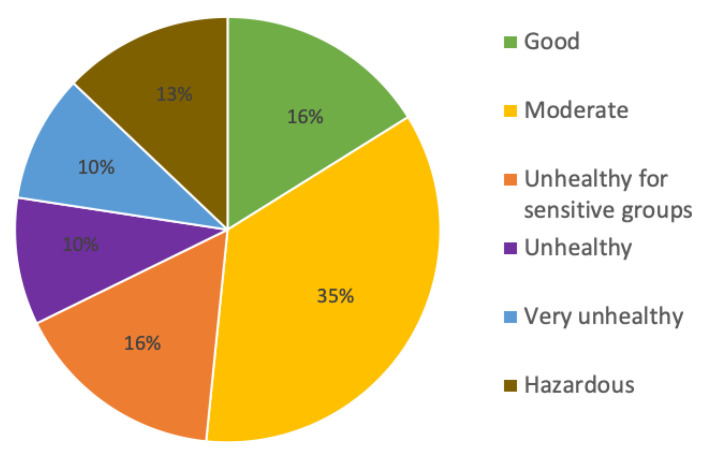
Contribution of Air Quality Index (AQI) during March 2021 in Gliwice.

**Figure 5 ijerph-19-14181-f005:**
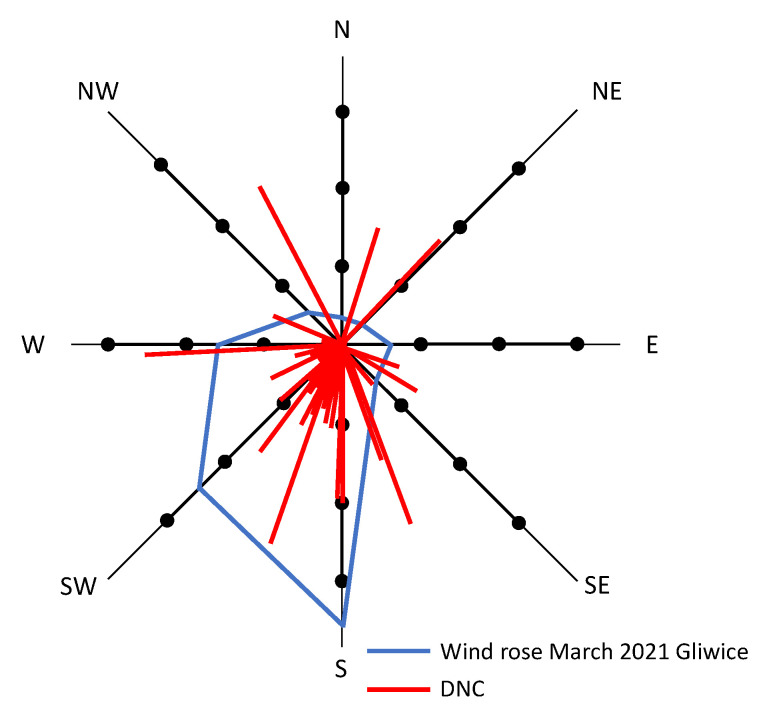
Wind rose vs. daily new cases (DNCs).

**Figure 6 ijerph-19-14181-f006:**
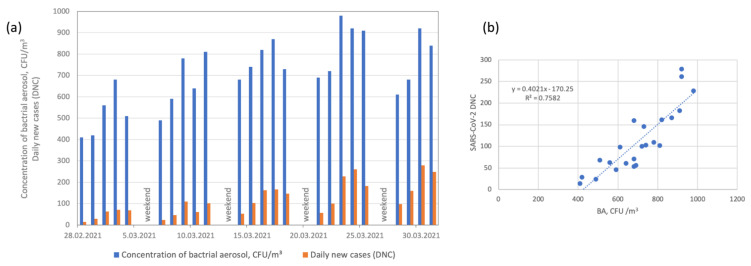
(**a**) Bacterial aerosol (BA) concentrations and SARS-CoV-2 daily new cases (DNCs). (**b**) Linear regression coefficients between bacterial aerosol (BA) concentrations and SARS-CoV-2 daily new cases (DNCs).

**Figure 7 ijerph-19-14181-f007:**
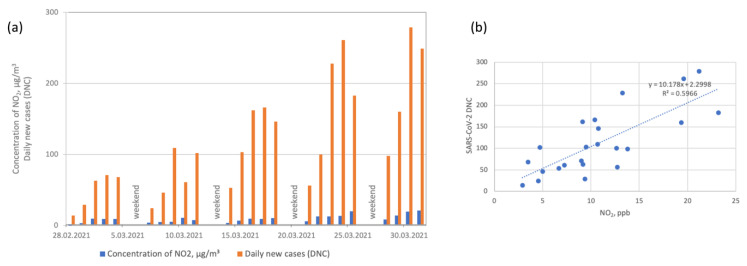
(**a**) NO_2_ concentrations and SARS-CoV-2 daily new cases (DNCs). (**b**) Linear regression coefficients between NO_2_ concentrations and SARS-CoV-2 daily new cases (DNCs).

**Table 1 ijerph-19-14181-t001:** Mean, median, minimum, and maximum values of parameters for daily new cases (DNCs) and all variables included in the analysis for March 2021.

Parameters	Mean	Median	SD	Min	Max	Lilliefors Test	W	Shapiro–Wilk Test
SARS-CoV-2 cases	112	100	74	14	279	*p* < 0.1	0.92	0.07
BAs, CFU/m^3^	703	690	161	410	980	*p* > 0.2	0.97	0.73
PM_2.5_ ^1^, μg/m^3^	33.1	24.2	20.9	10.9	76.1	*p* < 0.01	0.87	0.01
PM_10_ ^1^, μg/m^3^	45.3	36.6	25.8	17.2	102.9	*p* < 0.15	0.89	0.02
PM_10_ ^2^, μg/m^3^	38.9	28.1	26.5	11.4	106.0	*p* < 0.01	0.85	0.01
SO_2_, ppb	3.0	2.2	2.4	0.5	9.1	*p* < 0.01	0.79	<0.01
NO, ppb	5.0	4.3	3.6	1.5	15.5	*p* < 0.05	0.85	<0.01
NO_2_, ppb	10.8	9.5	5.6	2.9	23.2	*p* < 0.15	0.93	0.09
NO_x_, ppb	14.8	14.3	7.9	5.0	35.1	*p* < 0.2	0.91	0.05
O_3_, ppb	22.5	23.6	5.9	5.5	31.4	*p* > 0.2	0.94	0.15
CO, ppm	0.4	0.3	0.2	0.2	0.9	*p* < 0.05	0.86	<0.01
t, °C	4.4	3.4	3.9	−0.3	13.3	*p* < 0.15	0.91	0.03
RH, %	73.3	71.5	6.9	61.5	88.8	*p* > 0.2	0.97	0.58
P, hPa	996.9	995.6	7.4	985.9	1013.4	*p* < 0.1	0.94	0.17
Wind speed, m/s	1.5	1.3	0.7	0.5	2.8	*p* > 0.2	0.95	0.34
Wind direction, °	185.8	195.4	69.4	17.8	324.5	*p* < 0.05	0.94	0.19

^1^ Monitoring station. ^2^ Mobile air quality station at the sampling site.

**Table 2 ijerph-19-14181-t002:** Correlation matrix for SARS-CoV-2 daily new cases (DNCs), bacterial aerosol concentrations, and ambient air pollution levels during March 2021.

Parameters	DNCs	BAs	PM_2.5_ ^1^	PM_10_ ^1^	PM_10_ ^2^	SO_2_	NO	NO_2_	NO_x_	O_3_	CO
DNCs	1	**0.903**	−0.060	0.062	−0.205	0.377	**0.477**	**0.724**	**0.595**	−0.325	0.266
BA CFU/m^3^		1	−0.014	<0.01	−0.237	0.384	0.319	**0.632**	**0.479**	−0.247	0.207
PM_2.5_ ^1^ μg/m^3^			1	**0.869**	**0.891**	**0.513**	0.086	0.072	−0.007	−0.344	**0.614**
PM_10_ ^1^ μg/m^3^				1	**0.868**	**0.708**	0.329	0.320	0.245	**−** **0.491**	**0.828**
PM_10_ ^2^ μg/m^3^					1	**0.527**	−0.244	−0.005	−0.324	−0.226	**0.602**
SO_2_						1	**0.460**	**0.622**	**0.556**	**−** **0.517**	**0.776**
NO							1	**0.853**	**0.925**	**−** **0.809**	**0.506**
NO_2_								1	**0.970**	**−** **0.607**	**0.521**
NO_x_									1	**−** **0.681**	**0.465**
O_3_										1	**−** **0.637**

Correlation coefficients with *p* < 0.05 are in bold. ^1^ Monitoring station. ^2^ Mobile air monitoring lab at the sampling site.

**Table 3 ijerph-19-14181-t003:** Correlation matrix for SARS-CoV-2 daily new cases (DNCs), bacterial aerosol (BA) concentrations, and meteorological conditions during March 2021.

Parameters	DNCs	BAs	Temperature	RH	Atmospheric Pressure	Wind Speed	Wind Direction
Temperature	0.387	0.251	1	−0.309	0.278	−0.089	−0.024
RH	−0.377	−0.386	−0.309	1	0.007	0.194	0.409
Atmospheric pressure	−0.148	**−** **0.426**	0.278	0.007	1	−0.367	−0.010
Wind speed	0.016	0.041	−0.089	0.194	−0.367	1	0.005
Wind direction	**−** **0.477**	−0.288	0.024	0.409	−0.010	0.005	1

Correlation coefficients with *p* < 0.05 are in bold.

**Table 4 ijerph-19-14181-t004:** Bacterial species identifications.

Species of Isolated Bacteria
*Bacillus cereus*
*Bacillus subtilis*
*Bacillus flexus*
*Bacillus licheniformis* *Paenibacillus barengoltzii*
*Micrococcus luteus*
*Macrococcus equipercicus*
*Macrococcus brunensis*
*Nocardia alba*
*Lactobacillus crispatus*

## Data Availability

All data used in the paper are publicly available. We may supply the data that we gathered from public sources upon the making of a reasonable request to the corresponding author.

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
