# Peer review of "Impact of Different Air Pollutants (PM10, PM2.5, NO2, and Bacterial Aerosols) on COVID-19 Cases in Gliwice, Southern Poland"

_ijerph, 2022, doi:10.3390/ijerph192114181_

Round 1

Reviewer 1 Report

Comments

A very interesting scientific study of air pollution impact on Covid -19 cases.

The article use the English language correctly. Nevertheless the article needs some corrections and in my opinion the conclusions are incomplete and need re-writing.

More specific my comments are:

·         Please give information about the exact time period of the available data used in the specific study in section 2.2.

·         It seems that there is a problem with the axis. At figures 2 and 3 at the end of the study period, in figure 2, the DNC cases approach and even exceed the 250 cases but in figure 3 the DNC cases are below 250! Please correct it.

·         In figure 6a Bacterial aerosol (BA) concentration and SARS-CoV-2 daily new cases (DNC) are in the same axis with values > 400. This doesn’t follow the given information from Figures 2, 3.

Lines -54-55: Please give bibliography

Lines 87-88: NOx, include NO and NO2

Line 88: better “meteorological” parameters

Line 91: “mobile air monitoring laboratory for air pollutant emission measurements”. It would be better to be replaced with: mobile air quality/ pollution monitoring station

Line: Please justify why the Lilliefors and the Shapiro-Wilk tests were chosen and not others.

Lines 148-149: Please explain the meaning of “other results”. Are these results of the same time period or it is a general conclusion? Figures 2 and 3 present the variability of DNC and PM but a case of death is something complicated/ different and has to do with the severity of the virus to each organism, the status of the human body (underlying diseases), the medical protocol that follow at the hospital admissions etc. A death may come days or months affect the infection.

Lines 158-161: Could you explain why this is happening? Has to do, maybe, with ventilation, stop of heating season or changes of the virus severity (milder mutations of the virus)?

Line 179: A 0,477 (<0,7) Correlation coefficient means no strong correlation.

It would be interesting to present the Correlation matrix between the SARS-CoV-2 daily new cases (DNC) bacterial aerosol concentrations and ambient air pollution during 23/11/21-28/2/21 in order to make comparisons.

Please explain the results of the wind roses in figures 5. It seems that the S and SW sectors are of great importance. Can you explain the results of the right wind rose?

In figure 6a there is only one y axis.

I hope my comments to be helpful for the article edition.

Author Response

October 23rd, 2022

Manuscript ID: ijerph-1966409

Title: “Impact of different air pollutants (PM10, PM2.5, NO2, and bacterial aerosol) on COVID-19 cases in Gliwice, Southern Poland "

Response to the comments of Reviewer

We appreciate your response and the helpful suggestions for improving the presentation of our study. Below please find details on all modifications as they relate to your comments on the original manuscript.

Reviewer 1

A very interesting scientific study of air pollution impact on Covid -19 cases. The article use the English language correctly. Nevertheless the article needs some corrections and in my opinion the conclusions are incomplete and need re-writing.

More specific my comments are:

Ad.1

Please give information about the exact time period of the available data used in the specific study in section 2.2.

Thank You. This has now been corrected.

Ad. 2

It seems that there is a problem with the axis. At figures 2 and 3 at the end of the study period, in figure 2, the DNC cases approach and even exceed the 250 cases but in figure 3 the DNC cases are below 250! Please correct it.

Thank You for this comment. As the PM and DNC are 24h averages the graph has been changed into column and it has been changed in the revised manuscript.

Ad.3

In figure 6a Bacterial aerosol (BA) concentration and SARS-CoV-2 daily new cases (DNC) are in the same axis with values > 400. This doesn’t follow the given information from Figures 2, 3.

It has been changed in the revised manuscript.

Ad.4

Lines -54-55: Please give bibliography

The following supporting references have been included:

Albano, G.D.; Montalbano, A.M.; Gagliardo, R.; Anzalone, G.; Profita, M. Impact of Air Pollution in Airway Diseases: Role of the Epithelial Cells (Cell Models and Biomarkers). Int J Mol Sci 2022, 23.

Ciencewicki, J.; Jaspers, I. Air Pollution and Respiratory Viral Infection. Inhal Toxicol 2007, 19, 1135–1146, doi:10.1080/08958370701665434.

Van der Valk, J.P.M.; In ’t Veen, J.C.C.M. The Interplay Between Air Pollution and Coronavirus Disease (COVID-19). J Occup Environ Med 2021, 63

Ad.5

Lines 87-88: NOx, include NO and NO2

It has been corrected in the revised manuscript.

Ad.6

Line 88: better “meteorological” parameters

It has been added in the revised manuscript.

Ad.7

Line 91: “mobile air monitoring laboratory for air pollutant emission measurements”. It would be better to be replaced with: mobile air quality/ pollution monitoring station.

It has been changed in the revised manuscript.

Ad.8

Line: Please justify why the Lilliefors and the Shapiro-Wilk tests were chosen and not others.

We chose these specific statistical tests because they are used to evaluate samples containing less than 100 observations, as in our studies.

Ad.9

Lines 148-149: Please explain the meaning of “other results”. Are these results of the same time period or it is a general conclusion?

Dear Reviewer, other results presented in the publications listed below were not from the precisely same period as ours but were also obtained during the Covid-19 pandemic. The research results show a general conclusion that poorer air quality is linked with increased incidence of SARS-COV-2 cases.

  • Measurements carried out from March 2020 to June 2021

Semczuk-Kaczmarek, K.; Rys-Czaporowska, A.; Sierdzinski, J.; Kaczmarek, L.D.; Szymanski, F.M.; Platek, A.E. Association between Air Pollution and COVID-19 Mortality and Morbidity. Intern Emerg Med 2021, doi:10.1007/s11739-021-02834-5.

  • Measurements carried out from January 2020 to April 2020

Jiang, Y.; Xu, J. The Association between COVID-19 Deaths and Short-Term Ambient Air Pollution/Meteorological Condition Exposure: A Retrospective Study from Wuhan, China. Air Qual Atmos Health 2021, 14, doi:10.1007/s11869-020-00906-7.

  • Measurements carried out from January 2020 to March 2020

Comunian, S.; Dongo, D.; Milani, C.; Palestini, P. Air Pollution and Covid-19: The Role of Particulate Matter in the Spread and Increase of Covid-19’s Morbidity and Mortality. Int J Environ Res Public Health 2020, 17, 1–22, doi:10.3390/ijerph17124487.

  • Measurements carried out from March 2021 to June 2021

Meo, S.A.; Almutairi, F.J.; Abukhalaf, A.A.; Alessa, O.M.; Al-Khlaiwi, T.; Meo, A.S. Sandstorm and Its Effect on Particulate Matter PM 2.5, Carbon Monoxide, Nitrogen Dioxide, Ozone Pollutants and SARS-CoV-2 Cases and Deaths. Science of the Total Environment 2021, 795, doi:10.1016/j.scitotenv.2021.148764.

Ad.10

Figures 2 and 3 present the variability of DNC and PM but a case of death is something complicated/ different and has to do with the severity of the virus to each organism, the status of the human body (underlying diseases), the medical protocol that follow at the hospital admissions etc. A death may come days or months affect the infection.

We fully agree with the Reviewer, but Figures 2 and 3 show an increase in the number of daily new cases (DNC) of disease related to particulate pollution, without concluding about the number of deaths.

Ad.11

Lines 158-161: Could you explain why this is happening? Has to do, maybe, with ventilation, stop of heating season or changes of the virus severity (milder mutations of the virus)?

Most of the air pollution across Poland is the result of the country’s dependence on coal to power its homes and economy. The country’s coal industry remains an important part of the local economy. And household heating is the main contributor to particulate pollution. This source of pollution rises dramatically in winter, and hence, at that time, it could have contributed to the increased number of COVID-19 cases. The relationship between PM and DNC (daily new cases) has been losing importance since March (spring season, end of the heating season) despite the continuous increase in the number of SARS-CoV-2 cases. In the spring season in Poland, we note high concentrations of bacterial aerosol (BA) that may have contributed to the increase in the number of DNC, but this phenomenon requires additional analysis.

Ad.12

Line 179: A 0,477 (<0,7) Correlation coefficient means no strong correlation.

The fragment has been corrected into:

DNC are significantly correlated with bacterial aerosols (BA) and NO2. Correlation coefficients (r) were 0.903 and 0.724, respectively.

Ad.13

It would be interesting to present the Correlation matrix between the SARS-CoV-2 daily new cases (DNC) bacterial aerosol concentrations and ambient air pollution during 23/11/21-28/2/21 in order to make comparisons.

We agree with the Reviewer’s comment however, we do not have accurate data from this period.

Ad.14

Please explain the results of the wind roses in figures 5. It seems that the S and SW sectors are of great importance. Can you explain the results of the right wind rose?

We have deleted the annual wind roses from the region and commented the only one from the study period and area as follows:

“Figure 5 presents wind rose diagram from March 2021 monitored by mobile air quality station located in Gliwice. The result of study indicates that the wind in study area dominantly blows towards South and Southwest direction and the wind speed values were low in range from 0.5 to 2.8 m/s (Table 1). As it can be seen the highest COVID-19 cases fit in with wind direction blows”.

Ad.15

In figure 6a there is only one y axis.

It has been corrected in the revised manuscript.

I hope my comments to be helpful for the article edition.

Once again, we would like to express our appreciation to You for Your efforts and helpful comment.

Yours sincerely,

Anna Mainka and Ewa BrÄ…goszewska

Reviewer 2 Report

This manuscript described the relationship between COVID-19 daily new cases and air pollution. Additionally, the relationship between COVID-19 daily new cases and bacterial aerosol concentrations are analyzed. The conclusions of this manuscript can improve the knowledge about the link between air pollution and COVID-19. In this sense, this paper has some scientific significance and the important application value. However, the analysis method of this manuscript needs to be improved for more scientific results. The authors should consider revising it.

Detailed comments:

(1) The acute health effects of short-term exposure to air pollutants are mainly studied by time series studies, case-crossover studies or panel studies. The routine epidemiological approaches evaluate the impact of exposure to air pollution based on the regressed relative risk (morbidity or mortality) with time series from hospital clinic data, pollutant concentration data, meteorological data, and other data sources. One of the widely used methods in time-series studies is based on the generalized additive model (GAM) with Poisson regression. Considering the comprehensive effects of meteorological factors and air pollutants of COVID-19 daily new cases, the authors should supplement the results obtained by these important statistical analysis methods to enhance the understanding of health risks.

(2) The authors suggest that the bacterial infection may cause an increase in COVID-19 patients. The insightful discussion and researching significance on the relationship between the seasonal patterns of bioaerosols and COVID-19 daily new cases are lacked in the whole paper. The environmental significance of bioaerosols is not clarified in the introduction and discussion sections of this paper. Please supplement relevant discussion and analysis.

Author Response

October 23rd, 2022

Manuscript ID: ijerph-1966409

Title: “Impact of different air pollutants (PM10, PM2.5, NO2, and bacterial aerosol) on COVID-19 cases in Gliwice, Southern Poland "

Response to the comments of Reviewer

We appreciate your response and the helpful suggestions for improving the presentation of our study. Below please find details on all modifications as they relate to your comments on the original manuscript.

Reviewer 2

This manuscript described the relationship between COVID-19 daily new cases and air pollution. Additionally, the relationship between COVID-19 daily new cases and bacterial aerosol concentrations are analyzed. The conclusions of this manuscript can improve the knowledge about the link between air pollution and COVID-19. In this sense, this paper has some scientific significance and the important application value. However, the analysis method of this manuscript needs to be improved for more scientific results. The authors should consider revising it.

Detailed comments:

(1) The acute health effects of short-term exposure to air pollutants are mainly studied by time series studies, case-crossover studies or panel studies. The routine epidemiological approaches evaluate the impact of exposure to air pollution based on the regressed relative risk (morbidity or mortality) with time series from hospital clinic data, pollutant concentration data, meteorological data, and other data sources. One of the widely used methods in time-series studies is based on the generalized additive model (GAM) with Poisson regression. Considering the comprehensive effects of meteorological factors and air pollutants of COVID-19 daily new cases, the authors should supplement the results obtained by these important statistical analysis methods to enhance the understanding of health risks.

We agree with the Reviewer that the use of the generalized additive model (GAM) with Poisson regression would be a good idea as an additional point to present our research results. However, in our opinion, these methods of statistical analysis could form the basis for another publication. Of course, if in Your opinion the addition of GAM model is required in the article to be accepted for the publication in IJERPH, we will do so. However, we will need additional time to prepared and the additional author will be included.

(2) The authors suggest that the bacterial infection may cause an increase in COVID-19 patients. The insightful discussion and researching significance on the relationship between the seasonal patterns of bioaerosols and COVID-19 daily new cases are lacked in the whole paper. The environmental significance of bioaerosols is not clarified in the introduction and discussion sections of this paper. Please supplement relevant discussion and analysis.

Dear Reviewer, we really appreciate Your pointing this out, but an insightful discussion of the topic of the meaning of the relationship between the seasonal patterns of bioaerosols and COVID-19 daily new cases is impossible due to the lack of literature data in this regard.

We added proper information into the text:

Worldwide studies have revealed that BA concentrations vary among different types of outdoor environments, with considerable seasonal variations as well (Haas et al. 2010; BrÄ…goszewska and Pastuszka 2018; Woo et al. 2013). BA concentration and composition in the outdoor air can be influenced by specific micro- and macroscale determinants, such as land use, emission sources, air humidity, temperature, and UV radiation (BrÄ…goszewska and Pastuszka 2018; Ruiz-Gil et al. 2020; Šantl-Temkiv et al. 2018).

Bacterial aerosol particles play an important role as health risk factors, and exposure to these particles is associated with a varied range of health effects, including three major groups: infections, toxic reactions, and allergic reactions (Chegini et al. 2020; BrÄ…goszewska and BiedroÅ„ 2021). Therefore, the main aim of our study is to present the impact of bacterial aerosol present in the ambient air on the increase in COVID-19 cases in Gliwice in the Upper Silesia region of Poland, which is one of the most polluted areas in the EU (BrÄ…goszewska, and BiedroÅ„ 2021).”

BrÄ…goszewska, Ewa, and Izabela BiedroÅ„. 2021. “Efficiency of Air Purifiers at Removing Air Pollutants in Educational Facilities: A Preliminary Study.” Frontiers in Environmental Science 9 (September): 370. https://doi.org/10.3389/fenvs.2021.709718.

BrÄ…goszewska, Ewa, and Józef S. Pastuszka. 2018. “Influence of Meteorological Factors on the Level and Characteristics of Culturable Bacteria in the Air in Gliwice, Upper Silesia (Poland).” Aerobiologia 34 (2): 241–55. https://doi.org/10.1007/s10453-018-9510-1.

Chegini, Farhad Mirkhond, Abbas Norouzian Baghani, Mohammad Sadegh Hassanvand, Armin Sorooshian, Somayeh Golbaz, Rounak Bakhtiari, Asieh Ashouri, Mohammad Naimi Joubani, and Mahmood Alimohammadi. 2020. “Indoor and Outdoor Airborne Bacterial and Fungal Air Quality in Kindergartens: Seasonal Distribution, Genera, Levels, and Factors Influencing Their Concentration.” Building and Environment 175 (May): 106690. https://doi.org/10.1016/j.buildenv.2020.106690.

Haas, Doris, Martina Unteregger, Juliana Habib, Herbert Galler, Egon Marth, and Franz F. Reinthaler. 2010. “Exposure to Bioaerosol from Sewage Systems.” Water, Air, and Soil Pollution 207 (1–4): 49–56. https://doi.org/10.1007/s11270-009-0118-5.

Ruiz-Gil, Tay, Jacquelinne J. Acuña, So Fujiyoshi, Daisuke Tanaka, Jun Noda, Fumito Maruyama, and Milko A. Jorquera. 2020. “Airborne Bacterial Communities of Outdoor Environments and Their Associated Influencing Factors.” Environment International. https://doi.org/10.1016/j.envint.2020.106156.

Šantl-Temkiv, Tina, Ulrich Gosewinkel, Piotr Starnawski, Mark Lever, and Kai Finster. 2018. “Aeolian Dispersal of Bacteria in Southwest Greenland: Their Sources, Abundance, Diversity and Physiological States.” FEMS Microbiology Ecology 94 (4). https://doi.org/10.1093/femsec/fiy031.

Woo, Anthony C., Manreetpal S. Brar, Yuki Chan, Maggie C Y Lau, Frederick C C Leung, James A. Scott, Lilian L P Vrijmoed, Peyman Zawar-Reza, and Stephen B. Pointing. 2013. “Temporal Variation in Airborne Microbial Populations and Microbially-Derived Allergens in a Tropical Urban Landscape.” Atmospheric Environment 74: 291–300. https://doi.org/10.1016/j.atmosenv.2013.03.047.

Once again, we would like to express our appreciation to You for Your efforts and helpful comment.

Yours sincerely,

Anna Mainka and Ewa BrÄ…goszewska

Reviewer 3 Report

In the manuscript entitled "Impact of different air pollutants (PM10, PM2.5, NO2, and bac-2 terial aerosol) on COVID-19 cases in Gliwice, Southern Poland" authors tried to evaluate the influence of meteorological and air pollutants on the number of daily active COVID-19 cases in Gliwice. Though there are plenty of articles published on the similar analysis over different regions of the world, the analysis showing the possible link between bacterial aerosol (BA) and daily COVID cases makes this article worth publishing. However, there are plenty of typographical and grammatical erros through out the manuscript that need to be thoroughly corrected. I strongly suggest the authors to seek help of the professional english editor to improve the readability of this manuscript before getting accepted for publication. Few of the typo errors are listed here, still there are many more that is difficult to list all.

Line 64:remove "for example" repeated

Line 66: full stop missing after "analysed [19–23]"

Line 68: "study was" can be "study is"

Line 90: "PM10, as well as all gaseous and meteorological parameters," commas can be removed

Line 91: imission spelt wrong

Line 94: "β-radiation meter" the full form of BAM is "Beta Attenuation Monitor"

Line 104: "Air pollutants levels" should be "Air pollutant levels"

Line 108:  Insert comma after "step"

Line 114: change "numbers of cases" to "number of cases"

Line 122: insert comma after "distributed"

Line125: change "as well" to "as well as"

Line 167:insert comma after "guidelines [43]"

Line 177:insert comma after "suggests that"

Figure 6: "ug/m3" missplet in fig6a- y axis 

Author Response

October 23rd, 2022

Manuscript ID: ijerph-1966409

Title: “Impact of different air pollutants (PM10, PM2.5, NO2, and bacterial aerosol) on COVID-19 cases in Gliwice, Southern Poland "

Response to the comments of Reviewer

We appreciate your response and the helpful suggestions for improving the presentation of our study. Below please find details on all modifications as they relate to your comments on the original manuscript.

Reviewer 3

In the manuscript entitled "Impact of different air pollutants (PM10, PM2.5, NO2, and bacterial aerosol) on COVID-19 cases in Gliwice, Southern Poland" authors tried to evaluate the influence of meteorological and air pollutants on the number of daily active COVID-19 cases in Gliwice. Though there are plenty of articles published on the similar analysis over different regions of the world, the analysis showing the possible link between bacterial aerosol (BA) and daily COVID cases makes this article worth publishing. However, there are plenty of typographical and grammatical erros through out the manuscript that need to be thoroughly corrected. I strongly suggest the authors to seek help of the professional english editor to improve the readability of this manuscript before getting accepted for publication. Few of the typo errors are listed here, still there are many more that is difficult to list all.

Line 64: remove "for example" repeated

Line 66: full stop missing after "analysed [19–23]"

Line 68: "study was" can be "study is"

Line 90: "PM10, as well as all gaseous and meteorological parameters," commas can be removed

Line 91: imission spelt wrong

Line 94: "β-radiation meter" the full form of BAM is "Beta Attenuation Monitor"

Line 104: "Air pollutants levels" should be "Air pollutant levels"

Line 108:  Insert comma after "step"

Line 114: change "numbers of cases" to "number of cases"

Line 122: insert comma after "distributed"

Line125: change "as well" to "as well as"

Line 167:insert comma after "guidelines [43]"

Line 177:insert comma after "suggests that"

Figure 6: "ug/m3" missplet in fig6a- y axis - Thank You for pointing this out. We changed the unit. In the revised version it is Figure 7.

Dear Reviewer, thank you for these remarks. The typographical and grammatical errors have already been corrected in the revised manuscript.

Once again, we would like to express our appreciation to You for Your efforts and helpful comment.

Yours sincerely,

Anna Mainka and Ewa BrÄ…goszewska
